# Functional role of formate dehydrogenase 1 (FDH1) for host and nonhost disease resistance against bacterial pathogens

Seonghee Lee[1,2]*, Ramu S. Vemanna[1¤a], Sunhee Oh[1], Clemencia M. Rojas[1¤b], Youngjae Oh[2], Amita Kaundal[1¤c], Taegun Kwon[1¤d], Hee-Kyung Lee[1], Muthappa Senthil-Kumar[1¤e], Kirankumar S. Mysore[1,3,4]*

1 Noble Research Institute, LLC, Ardmore, OK, United States of America, 2 Gulf Coast Research and Education Center, Institute of Food and Agricultural Science, University of Florida, Wimauma, FL, United States of America, 3 Institute for Agricultural Biosciences, Oklahoma State University, Ardmore, OK, United States of America, 4 Department of Biochemistry and Molecular Biology, Oklahoma State University, Stillwater, OK, United States of America

¤a Current address: Laboratory of Plant Functional Genomics, Regional Center for Biotechnology, Faridabad, Haryana, India
¤b Current address: Department of Entomology and Plant Pathology, University of Arkansas, Fayetteville, AR, United States of America
¤c Current address: Department of Plants, Soils and Climate, Utah State University, Logan, Utah, United States of America
¤d Current address: Genomics Center, University of North Texas, Denton, TX, United States of America
¤e Current address: National Institute of Plant Genome Research, New Delhi, India
* seonghee105@ufl.edu (SL); kmysore@okstate.edu (KSM)

**Data Availability Statement:** All relevant data are within the manuscript and its Supporting Information files.

## Abstract

Nonhost disease resistance is the most common type of plant defense mechanism against potential pathogens. In the present study, the metabolic enzyme formate dehydrogenase 1 (FDH1) was identified to associate with nonhost disease resistance in *Nicotiana benthamiana* and *Arabidopsis thaliana*. In Arabidopsis, *AtFDH1* was highly upregulated in response to both host and nonhost bacterial pathogens. The *Atfdh1* mutants were compromised in nonhost resistance, basal resistance, and gene-for-gene resistance. The expression patterns of salicylic acid (SA) and jasmonic acid (JA) marker genes after pathogen infections in *Atfdh1* mutant indicated that both SA and JA are involved in the *FDH1*-mediated plant defense response to both host and nonhost bacterial pathogens. Previous studies reported that FDH1 localizes to mitochondria, or both mitochondria and chloroplasts. Our results showed that the AtFDH1 mainly localized to mitochondria, and the expression level of FDH1 was drastically increased upon infection with host or nonhost pathogens. Furthermore, we identified the potential co-localization of mitochondria expressing FDH1 with chloroplasts after the infection with nonhost pathogens in Arabidopsis. This finding suggests the possible role of FDH1 in mitochondria and chloroplasts during defense responses against bacterial pathogens in plants.

**Funding:** This study received support from the Noble Research Institute, LLC (https://www.noble.org/about/) in the form of salaries for authors SL, RV, SO, CR, AK, TK, HKL, MSK, and KM. The specific roles of these authors are articulated in the 'author contributions' section. The funders had no role in study design, data collection and analysis, decision to publish, or preparation of the manuscript. No additional external funding was received for this study.

**Competing interests:** The authors of the study have read the journal's policies and have the following competing interests to declare: At the time that the study was conducted, authors SL, RV, SO, CR, AK, TK, HKL, MSK, and KM were paid employees of the Noble Research Institute, LLC (https://www.noble.org/about/). This affiliation does not alter our adherence to PLOS ONE policies on sharing data and materials. There are no patents, products in development or marketed products associated with this research to declare. The authors, [SL, RV, SO, CR, AK, TK, HKL, MSK, KM], were affiliated Noble Research during the research project period. However, no more authors listed in the manuscript are currently working at the Noble Research. This affiliation does not alter our adherence to PLOS ONE policies on sharing data and materials.

# Introduction

Nonhost resistance provides basic protection to plants and is also the most durable form of resistance to the majority of potential pathogens [1–3]. In general, both basal and nonhost resistance are controlled by quantitative trait loci (QTL). Disease resistance traits conferred by these QTLs have been widely used for developing new varieties for disease resistance [3–8]. In addition to QTLs, a number of studies have identified major plant genes involved in nonhost resistance against fungal and bacterial pathogens [2, 3, 6, 8–10]. However, the mechanism of nonhost resistance is not fully understood. Nonhost resistance against bacterial pathogens can be broadly classified as two types; type I (no visible hypersensitive response [HR] cell death) and type II (HR cell death) nonhost resistances [2]. The efficacy of nonhost disease resistance could be based on the recognition of pathogen-associated molecular patterns (PAMPs) and/or pathogen effectors. A number of studies have showed that genes associated with nonhost resistance is often involved in basal defense mechanism (pre- or post-invasive defense). For instance, stomatal innate immunity is an important mechanism of pre-invasive defense. Proteins involved in post-invasive responses could activate HR-type cell death or ROS accumulation to develop nonhost resistance. Organelles such as mitochondria and chloroplast have been well described for their important role for the ROS-mediated programmed cell death. PAMPs are mainly perceived at the plasma membrane where the PAMP-triggered immunity (PTI) could be induced as the first defense barrier against various pathogens [11, 12]. One known PTI response is stomatal closure which is circumvented by the phytotoxin coronatine (COR) produced by the host pathogen *P. syringae* pv. tomato DC3000 [13]. COR has structural and functional similarity to jasmonates and jasmonic acid-isoleucine (JA-Ile), and contributes to the virulence of *P. syringae* pv. tomato DC3000 [14–16]. COR disrupts the accumulation of the plant defense hormone salicylic acid (SA) for stomatal reopening and bacterial propagation in both local and systemic tissues of *Arabidopsis* [17]. COR is also involved in promoting the entry of nonhost bacterial pathogens via stomata and nonhost bacterial growth at the initial stage of infection [18]. In addition to PTI, a number of pathogen effectors secreted into host cells can also induce another type of defense response referred to as effector-triggered immunity (ETI) [19, 20]. ETI is typically associated with resistance proteins belonging to the nucleotide-binding domain (NBD) and leucine-rich repeat-containing (NLR) family. ETI triggers a type of cell death known as the HR [20]. Despite the plant immune systems, compatible host bacterial pathogens in susceptible plants suppress both basal and nonhost resistance responses thus causing disease.

Formate dehydrogenase 1 (FDH1) is a nicotinamide adenine dinucleotide (NAD+)-dependent enzyme that catalyzes the NAD-linked oxidation of formate to carbon dioxide. As a component of one-carbon metabolism in plants, most FDHs play an important role in response to various stresses in higher plants [21–26]. A previous report has shown that FDH1 regulates programmed cell death (PCD) in pepper against bacterial pathogens [23]. There is contradictory information regarding the localization of FDH1 in plant cell. According to the study by Choi (2014), FDH1 localizes to mitochondria and plays a role in hypersensitive cell death and the defense signaling pathway against bacterial pathogens in pepper. Several other reports also suggest there is mitochondrial localization of FDH1 in tobacco [27, 28]. Interestingly, several reports have described that FDH1 targets not only mitochondria, but also chloroplasts for its biological function [29, 30]. Chloroplast and mitochondria are the major targets of plant pathogen effectors, and the effectors targeting of these organelles inhibits the production of defense molecules including reactive oxygen species (ROS) [31–33]. Chloroplasts play a major role in generating ROS and nitric oxide to trigger defense responses such as PCD and HR against bacterial pathogens [34–36]. Mitochondria and chloroplasts also have been reported as the initial

organelle to recognize bacterial effectors and trigger plant immunity against bacterial pathogens [37, 38]. In other studies, the co-localizations of mitochondria with chloroplasts has been well characterized [39–41]. The physical interactions between mitochondria and chloroplasts would provide the means to transfer genetic information directly to the organelle genome, as well as to mediate signaling transduction [42, 43]. However, how chloroplast and mitochondria are functionally integrated for bacterial disease resistance is not fully understood. Particularly, previous conflicting results regarding the cellular localizations of FDH1 may suggest possible roles of FDH1 in the chloroplast as well as mitochondria for bacterial disease resistance.

In the current study, we demonstrated the novel role of FDH1 in nonhost disease resistance in *Nicotiana benthamiana* and Arabidopsis. The cellular localization of FDH1 was confirmed to be mitochondria, but it was also found that the protein targets chloroplasts during the defense responses against host and nonhost bacterial pathogens. We speculate that FDH1 may coordinate mitochondria- and chloroplast-mediated defense responses against bacterial pathogens in plants.

## Materials and methods

### Plant materials

*Nicotiana benthamiana* plants were grown in 10-centimeter diameter round pots with BM7 soil (SUNGRO Horticulture Distribution, Inc., Bellevue WA) in the greenhouse using the conditions described in the previous study [44]. Plants grown four weeks were used for virus-induced gene silencing (VIGS) experiments as described below. The ecotype of *Arabidopsis thaliana*, Col-0, was used as wild-type. Arabidopsis T-DNA knockout mutants for *AtFDH1* gene (At5g14780), SALK_118644 and SALK_118548, were obtained from the Arabidopsis Biological Resource Center (Columbus, OH). To identify the homozygous knockout T-DNA mutant plants, seedlings grown from the SALK_118644 and SALK_118548 seeds and their progeny were harvested for PCR-based genotyping. Primers were designed from SALK T-DNA verification primer design (http://signal.salk.edu/tdnaprimers.2.html), and PCR was performed using REDExtract-N-Amp™ Tissue PCR Kit (Sigma-Aldrich, St. Louis, MO). All mutant plants were made homozygous for their respective T-DNA insertion, and seeds were harvested for further experiments. For seedling-flood inoculation (45), Arabidopsis plants were grown in ½ Murashige and Skoog (MS) agar medium plates at 25˚C under short day condition (12 h light).

### VIGS in *Nicotiana benthamiana*

VIGS in *N. benthamiana* was performed as described [2]. In brief, *Agrobacterium tumefaciens* GV2260 containing TRV1, TRV2::00, and TRV2::*NbFDH1* was grown overnight on LB medium containing antibiotics (rifampicin, 25; kanamycin, 50) at 28˚C. Bacterial cells were harvested and re-suspended in induction medium (10 mM MES, pH 5.5; 200 μM acetosyringone), and incubated at room temperature on an orbital shaker for 5 hrs. Bacterial cultures containing TRV1 and TRV2 were mixed in equal ratios ($OD_{600}$ = 1) and infiltrated into *N. benthamiana* leaves using a 1 ml needleless syringe. The infiltrated plants were maintained in a greenhouse and used for studies 15 to 21 days post-infiltration.

### Bacterial culture and inoculation

Bacterial pathogens, *Pseudomonas syringae* pv. tabaci (*Pstab*), *P. syringae* pv. tomato T1 (*Pst* T1), and *P. syringae* pv. maculicola (*Psm*) were grown in King's B (KB) medium at 28˚C

overnight. The bacterial culture was centrifuged at 5,000 rpm for 10 min, and the cell pellet was re-suspended in 5 ml sterilized distilled water. For the inoculation assays in *N. benthamiana*, bacterial vacuum infiltration was performed using the concentration of $1 \times 10^4$ CFU/ml for both *N. benthamiana* host (*Pstab*) and nonhost (*Pst* T1) pathogens. For the inoculation assays in Arabidopsis, host (*Psm*) and nonhost (*Pstab*) pathogens were used for the inoculation followed by the seedling flood-inoculation method [45, 46].

## Bacterial disease assay in *N. benthamiana* and Arabidopsis

For disease assays in *N. benthamiana*, bacterial suspensions of host and nonhost pathogens ($1 \times 10^5$ CFU/ml) were vacuum-infiltrated in both silenced and control plants 2-week after TRV infection. The fully expanded leaves were used for disease assays, and the inoculated plants were kept in a growth chamber at 20–22˚C. The number of bacterial cells in leaf apoplast were measured 1, 2, and 3 days after inoculation in *N. benthamiana*. The bacterial population at day 0 was estimated from leaves harvested 1 hr after inoculation. Two leaf discs (0.5 cm$^2$) from each leaf were collected in 1.5 ml centrifuge tube containing 100 ul of sterilized distilled water. Samples were homogenized and plated on KB agar medium for measuring colony-forming units (CFU) per cm$^2$ of leaf area. A total of three leaves were used for each experiment. To visualize bacterial colonization at infected sites in leaves, *GFPuv*-expressing *P. syringae* pv. tabaci and *P. syringae* pv. tomato T1 were vacuum infiltrated, and plants were examined under UV light 3 days after inoculation [47].

For disease assays in Arabidopsis, a flood inoculation method was used to infect Arabidopsis [45, 46]. Disease symptoms were observed 3 days after inoculation. For bacterial counting, leaves were surface-sterilized with 10% bleach for one min to eliminate epiphytic bacteria and then washed with sterile distilled water twice. The leaves were then homogenized in sterile distilled water, and serial dilutions were plated onto KB plates. Bacterial growth was evaluated in three independent experiments.

## FDH1 protein localization in *N. benthamiana* and Arabidopsis

The full-length sequence of *AtFDH1* with native promoter was cloned into pMDC107 for *GFP* expression (*AtFDH1-GFP*). Stable Arabidopsis transgenic lines for the expression of *AtFDH1-GFP* were developed by floral dip transformation [48]. The localization of AtFDH1-GFP in epidermal cells was determined under the confocal laser scanning microscope (NIKON, Japan).

To observe the localization of AtFDH1, Arabidopsis wild-type Col-0 and *AtFDH1-GFP* expressing (under the control of *AtFDH1* promoter) transgenic plants in Col-0 were grown in ½ MS media for four weeks, and *AtFDH1-GFP* expression in epidermal cells of Arabidopsis was visualized using a confocal laser scanning microscope (NIKON, Japan). The leaf tissues were floated with the bacterial suspension of host pathogen *P. syringae* pv. maculicola ($1 \times 10^6$ CFU/ml) and nonhost pathogen *P. syringae* pv. tabaci ($1 \times 10^6$ CFU/ml). After one hour inoculation, the leaf tissues were washed with distilled water, and localization of FDH1-GFP was observed. For wounding stress, the adaxial epidermal peels from wild-type Col-0 and *AtFDH1-GFP* expressing transgenic plants were prepared in the MES buffer (10 mM, pH 6.5), and localization of AtFDH1 was imaged under the confocal laser scanning microscope (NIKON, Japan).

## Isolation of chloroplast and mitochondria

Arabidopsis leaves (10 g) were homogenized in 100 ml of grinding buffer containing 50 mM HEPES (pH 8.0), 2 mM EDTA, 1 mM MgCl$_2$, 0.33 M sorbitol, and 0.5 g/L BSA by using a

motor-driven blender (WARING 51BL30, two 5 s bursts at maximum speed). The homogenate was filtered through 3 layers of miracloth (Sigma-Aldrich, St. Louis MO, USA). The cleared homogenate was centrifuged at 1,500 g for 10 min at 4°C. The supernatant was used for isolation of mitochondria, and the pellet was used for chloroplast extraction. For the isolation of chloroplast, the pellet was re-suspended in 3 ml of grinding buffer with a paint brush. The chloroplast suspension was then loaded on top of linear Percoll gradient (2 ml of 70% PBF-Percoll (v/v), 4 ml of 50% PBF-Percoll (v/v), and 4 ml of 40% PBF-Percoll (v/v)) and centrifuged at 16,000 g for 20 min at 4°C. The lower green bands were collected for intact chloroplasts with a glass pipette, washed twice with wash buffer (50 mM HEPES, pH 8.0, 2 mM EDTA, 1 mM MgCl$_2$, 0.33 M sorbitol), and centrifuged at 1,500 g for 10 min at 4°C. The supernatant was discarded and the washed chloroplast pellet was collected for chloroplast protein extraction.

For the isolation of mitochondria, the supernatant was centrifuged at 3,000 g for 5 min at 4°C. The supernatant was transferred into a fresh centrifuge tube and centrifuged at 18,000 g for 20 min at 4°C. The greenish mitochondrial pellet was re-suspended carefully in 1 ml wash buffer with a fine paint brush and adjusted the final volume to 4.8 ml. 1.2 ml of 100% Percoll (Sigma-Aldrich, St. Louis, MO) was added and the total 6 ml of mitochondria homogenate was then loaded on top of linear Percoll gradient (5ml of 80% PBF-Percoll (v/v), 5ml of 33% PBF-Percoll). The mitochondria homogenate was centrifuged at 18,000 g for 1 hr and greenish upper band was collected. Mitochondria was rinsed twice with 15 ml wash buffer and centrifuged at 18,000 g for 20 min at 4°C. The supernatant was removed and the pellet was saved for mitochondria protein extraction.

## Protein extraction from chloroplast and mitochondria

The mitochondrial and chloroplast proteins were isolated [49] in protein extraction buffer; 50 mM Tris-HCL, pH 7.5, 75 mM NaCl, 0.2% Triton X-100, 5 mM EDTA, 5 mM EGTA, 1 mM DTT, 100 uM MG132, 10 mM NaF, 2 mM Na2VO4, and 1% protease inhibitor cocktail (Sigma Aldrich, St. Louis, USA). The extracted proteins were quantified using Bradford method [50], and equal known concentrations were taken for the assay. Proteins were blotted on a polyvinylidene fluoride (PVDF) membrane and Cox II antibody (Agrisera, Sweden, cat no. AS04 053A) for mitochondria and Rubisco or RBCL (Abiocode, CA, USA, cat.no. R3352-2) for chloroplast was used as markers to confirm the proteins. GFP antisera (Miltenyl Biotec, San Diego, CA, USA cat. no. 130-091-833) was used to detect the FDH1 protein levels. The primary HRP-conjugated GFP antisera were diluted to 1:10,000 and visualized using ECL solution (GE Healthcare Bio-Sciences, Pittsburgh, USA) and protein gel blots were imaged. The raw image data for western blot analysis is show in (S6 Fig).

## Quantitative reverse transcription PCR (RT-qPCR) analysis

Total RNA was extracted from Arabidopsis leaves infiltrated with water (mock control), host pathogen (*P. syringae* pv. maculicola) and nonhost pathogen (*P. syringae* pv. tabaci), sampled at 0, 12 and 24 hrs post-inoculation (hpi). RNA samples were treated with DNAseI (Ambion, Austin, TX) and used for cDNA synthesis using SuperScript III reverse transcriptase (Invitrogen, Grand Island, NY, USA). The cDNA was diluted to 1:20 and used for RT-qPCR using Power SYBR Green PCR master mix (Applied Biosystems, Foster City, CA, USA) with an ABI Prism 7900 HT sequence detection system (Applied Biosystems, Foster City, CA, USA). Arabidopsis *Ubiquitin 5* (*UBQ5*) and *Elongation factor 1α* (*EF1α*) were used as internal controls to ensure an equal amount of cDNA in individual reactions. Average Cycle Threshold (Ct) values calculated using Sequence Detection Systems (version 2.2.2; Applied Biosystems) from

**A**

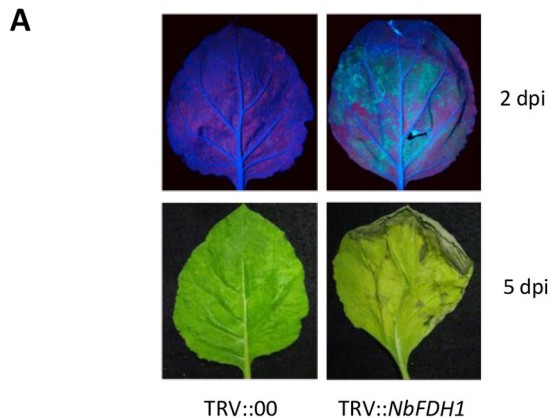

2 dpi

5 dpi

TRV::00          TRV::*NbFDH1*

**B**

*P. syringae* pv. tomato T1

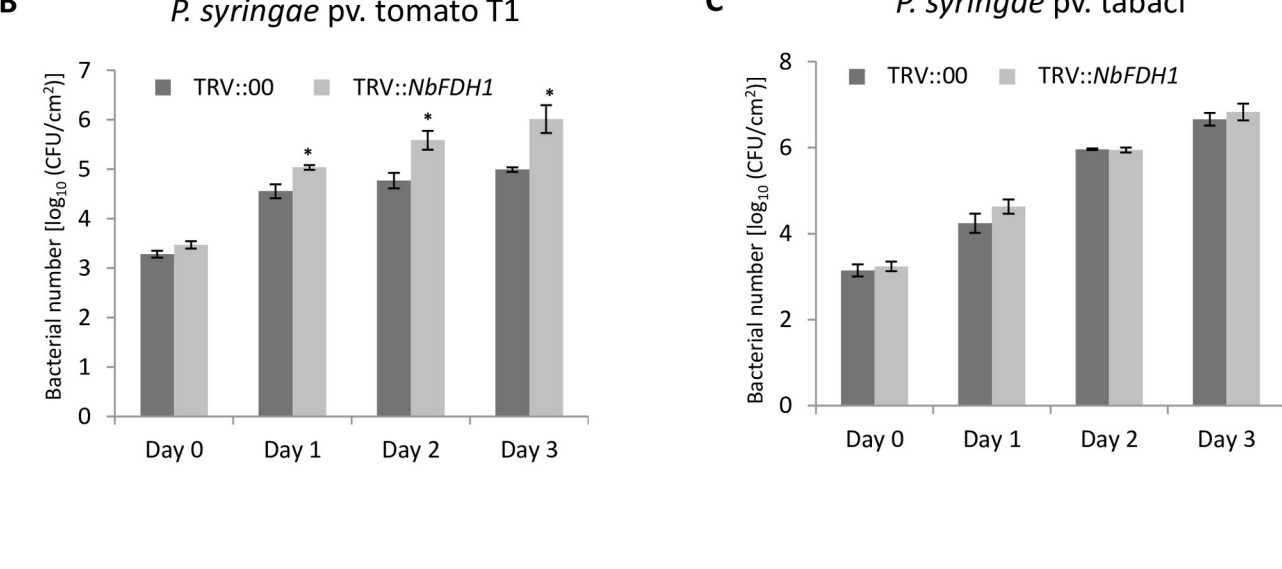

**C**

*P. syringae* pv. tabaci

**D**

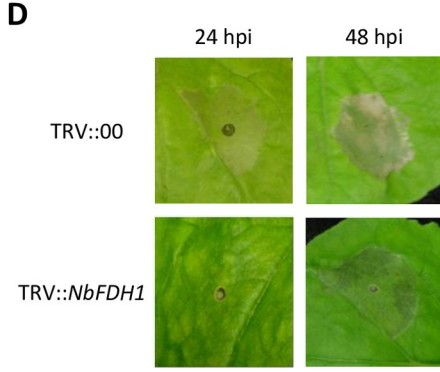

24 hpi          48 hpi

TRV::00

TRV::*NbFDH1*

**Fig 1. Virus-induced gene silencing of *NbFDH1* compromises nonhost resistance and elicitation of hypersensitive response in *N. benthamiana*.**
(A) GFP fluorescence associated with bacterial multiplication of nonhost bacteria in *NbFDH1* silenced *N. benthamiana* leaves. Two weeks old *N. benthamiana* seedlings were inoculated with TRV1 + TRV::00 (control) or TRV1 + TRV::*NbFDH1*. Three weeks after TRV inoculation, nonhost bacterial pathogen *P. syringae* pv. tomato T1 expressing *pDSK-GFP$_{uv}$* was vacuum infiltrated at $1\times10^4$ CFU/ml concentration. The photograph was taken under UV light 2 days post infection (dpi) as show in the upper panel. Visual disease symptoms were photographed at 5 dpi (lower panel). An increase in GFP fluorescence and disease symptoms were observed in TRV::*NbFDH1* inoculated but not in the TRV::00 inoculated plants. (B) and (C)

Bacterial titer of host and nonhost pathogens in both *NbFDH1*-silenced and control plants. TRV inoculated plants (described above) were vacuum inoculated with host (*P. syringae* pv. tabaci) or nonhost (*P. syringae* pv. tomato T1) bacterial pathogens ($1\times10^4$ CFU/ml), and bacteria were quantified by plating serial dilutions of leaf extracts. Asterisks indicate a significant difference from the control using Student's *t* test (P < 0.01). Bars represent mean, and error bars represent the standard deviation of three biological replicates (three technical replicates were used for each biological replicate). Each experiment showed similar results. (D) HR-related cell death in *NbFDH1*-silenced and control plants. High concentration ($1\times10^6$ CFU/ml) of nonhost pathogen *P. syringae* pv. tomato T1 was infiltrated using a needless syringe into fully expanded *N. benthamiana* leaves, three weeks after TRV inoculation. Cell death due to nonhost HR was observed and photographed 24 and 48 hpi.

duplicate samples and were used to determine the fold expression relative to controls. Two biological replicates of each sample and three technical replicates of each biological replicate were analyzed for RT-qPCR analysis.

## Results

### Formate dehydrogenase 1 is involved in nonhost disease resistance

Using virus-induced gene silencing (VIGS)-based forward genetics screening in *Nicotiana benthamiana*, we identified the clone 24E07 (NbME24E07) to be involved in nonhost disease resistance against the bacterial pathogen *Pseudomonas syringae* pv. tomato T1 [44, 51]. The cDNA insert in 24E07 clone was sequenced and BLAST results of the sequence showed that it was homologous to *NbFDH1*. Protein sequence analysis showed that NbFDH1 is 96% identical to SlFDH1 and 80% identical to AtFDH1 (S1 Fig). *FDH1* is a single copy gene in both monocot and dicot plants.

*Tobacco rattle virus* (TRV)-based VIGS of *NbFDH1* in *N. benthamiana* plants did not cause a visible phenotype regarding plant appearance. The downregulation of *NbFDH1* was about 70% in TRV::*NbFDH1* inoculated plants when compared to TRV:00 (non-silenced control) inoculated plants (S2 Fig). *NbFDH1*-silenced and non-silenced control plants were inoculated with host and nonhost pathogens. Upon vacuum infiltration with the nonhost pathogen *P. syringae* pv. tomato T1 containing *pDSK-GFPuv* [41] at $1\times10^4$ CFU/ml concentration, the bacteria multiplied more in *NbFDH1*-silenced plants than the non-silenced control as visualized by green fluorescence under UV light (Fig 1A). *NbFDH1* silenced plants showed necrotic disease symptoms in infected leaf tissues, while no disease symptoms were observed in the non-silenced control (Fig 1A). Further, the bacterial titer of nonhost pathogen *P. syringae* pv. tomato T1 was measured for three consecutive days after inoculation in both the *NbFDH1*-silenced and non-silenced control plants. Consistent with the disease symptoms and green fluorescence observed, *NbFDH1*-silenced plants had more bacterial titer compared to the non-silenced control (Fig 1B). In contrast to the nonhost pathogen, multiplication of the host pathogen *P. syringae* pv. tabaci was not different in *NbFDH1* silenced plants when compared to non-silenced control (Fig 1C).

To check if *NbFDH1* has a role in nonhost HR, *NbFDH1*-silenced and non-silenced control plants were syringe-infiltrated with a high level of inoculum ($1\times10^6$ CFU/ml) of the nonhost pathogen *P. syringae* pv. tomato T1. Non-silenced control showed a typical nonhost HR after 24 hpi whereas in *NbFDH1*-silenced lines, the HR was delayed until 48 hpi (Fig 1D). Together, these results suggest that *NbFDH1* plays a role in nonhost disease resistance against *P. syringae* pv. tomato T1 in *N. benthamiana*.

### Arabidopsis *fdh1* mutants show increased susceptibility to host-pathogen and nonhost pathogens

Two Arabidopsis T-DNA insertion mutants (Col-0 background) for *AtFDH1* gene (SALK118548: *Atfdh1-1* and SALK118644: *Atfdh1-3*) were identified in the Arabidopsis

A

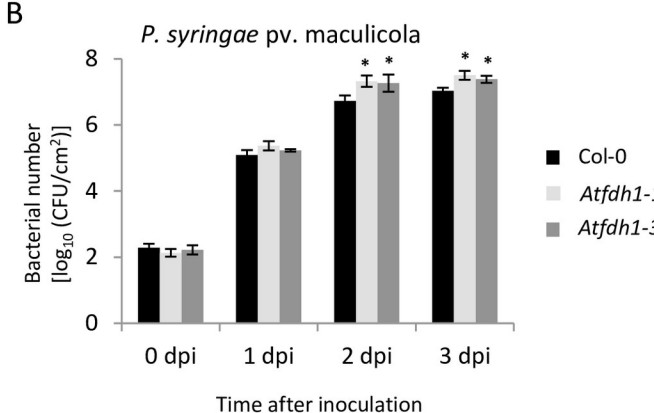

B

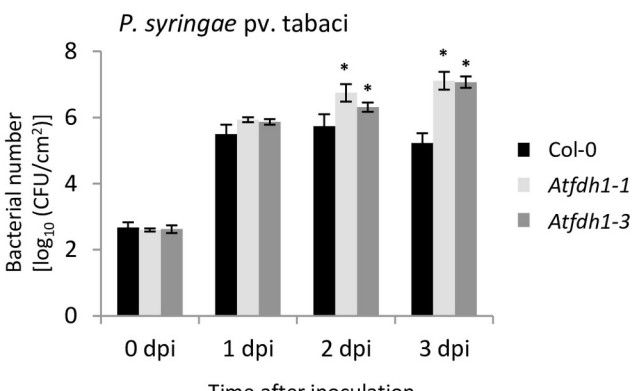

C

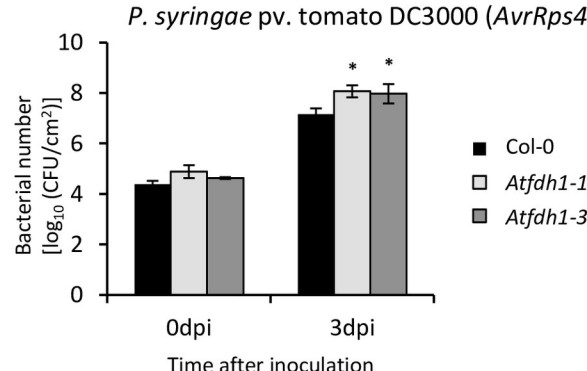

**Fig 2. Arabidopsis *Atfdh1* mutants are compromised in basal, nonhost, and gene-for-gene resistance.** (A) Disease symptoms of *Atfdh1-1* mutant after inoculation with host or nonhost pathogens. Two-week-old Arabidopsis wild-type (Col-0) and *Atfdh1-1* mutants grown in 1/2 strength MS medium under short-day conditions (8 hrs of daylight) were flood-inoculated with host (*P. syringae* pv. maculicola) or nonhost (*P. syringae* pv. tabaci) pathogens at $3 \times 10^6$ CFU/ml. Photographs were taken at four days post inoculation (dpi). (B) Bacterial titer of host and nonhost pathogens in *Atfdh1* mutants. Two-week-old Arabidopsis Col-0 and two *Atfdh1* mutant alleles (*Atfdh1-1* and *Atfdh1-3*) were flood-inoculated with host (*P. syringae* pv. maculicola) or nonhost (*P. syringae* pv. tabaci) pathogens at $1 \times 10^5$ CFU/ml. Bacterial titers at 0 to 3 dpi were measured by taking leaf disks from four inoculated plants for each line. (C) Quantification of host bacterial multiplication during gene-for-gene resistance. Leaves from 6-week-old plants of Col-0 and *Atfdh1* mutant alleles were syringe-infiltrated with avirulent (*P. syringae* pv. tomato DC3000 [*AvrRps4*]) bacterial strain at $2.8 \times 10^5$ CFU/ml concentration. Bacterial titer was measured at 0 and 3 dpi. Bars represent mean, and error bars represent standard deviation for four biological replicates with two independent experiments. The bacterial growth was similar between the experiments. Asterisks above bars represent statistically significant differences in comparison with wild-type plants using Student's *t*-test (P < 0.05).

T-DNA insertion lines and were obtained from the Arabidopsis Biological Resource Center. Homozygous T-DNA insertion lines were generated by selfing and confirmed by PCR. When wild-type (Col-0) and *Atfdh1* mutants were flood inoculated [45, 46] with the nonhost pathogen *P. syringae* pv. tabaci, *Atfdh1* mutants showed disease symptoms characterized by chlorosis at 5-day post inoculation (dpi), while Col-0 did not (Fig 2A). In addition, *Atfdh1* mutants had higher bacterial titer (approximately 18-fold) when compared to Col-0 plants at 3 dpi (Fig 2B). In response to an infection with a host pathogen, *P. syringae* pv. maculicola, both Col-0 and the *Atfdh1* mutants showed similar disease symptoms (Fig 2A). Interestingly, in contrast to the observation in *NbFDH1*-silenced *N. benthamiana* where the host pathogen titer did not differ between silenced and control plants, Arabidopsis host pathogen, *P. syringae* pv. maculicola, grew slightly more in the *Atfdh1* mutants when compared to Col-0 (Fig 2B).

To check if *AtFDH1* plays a role in gene-for-gene resistance, we infected Arabidopsis Col-0 plants that carry multiple resistance (*R*) genes, including *RPS4* with avirulent *P. syringae* pv. tomato DC3000 (*AvrRPS4*). After 3 dpi, *P. syringae* pv. tomato DC3000 (*AvrRPS4*) grew ~3 logs in wild-type Col-0, but a significantly higher growth of bacteria was observed in the *Atfdh1* mutant lines (Fig 2C). The delayed HR-associated cell death was also found in *NbFDH1*-silenced *N. benthamiana* plants (Fig 1D). These results suggest that *AtFDH1* confers plant defense associated with gene-for-gene resistance mechanisms.

## *AtFDH1* is induced in response to host and nonhost bacterial pathogens

In the publically available gene expression databases (TAIR), *AtFDH1* is strongly expressed after 24h of inoculation with the virulent pathogen *P. syringae* pv. tomato DC3000 and the avirulent pathogen *P. syringae* pv. tomato (*AvrRPM1*) (https://www.arabidopsis.org/servlets/TairObject?id=136173&type=locus; S3A Fig). This agrees with the previous study of mitochondrial *FDH1* in pepper [23]. We also found that *AtFDH1* gene expression is induced after host or nonhost pathogen inoculation (S3B Fig). After inoculation with the virulent pathogen *P. syringae* pv. maculicola, *FDH1* expression increased slightly (less than 0.5-fold) in comparison to mock-inoculated plants. Inoculation with the nonhost pathogen *P. syringae* pv. tabaci caused a higher induction of *FDH1* and its level of expression was about 2-fold higher than in mock-inoculated plants (S3B Fig). These results suggest that *FDH1* may play a greater role in nonhost disease resistance.

## Mutation of *AtFDH1* alters the SA-mediated defense hormonal pathway to bacterial pathogens

The gene expression of *AtFDH1* was induced in response to both host and nonhost pathogens (S3A and S3B Fig). To examine if the resistance mechanism was related to a known common defense pathway, such as salicylic acid (SA) and Jasmonic acid (JA), we conducted quantitative RT-PCR (RT-qPCR) for the expression of plant defense related genes in wild-type Col-0 and

the *Atfdh1* mutant without any pathogen inoculation and at 24 hpi with the host pathogen *P. syringae* pv. maculicola or the nonhost pathogen *P. syringae* pv. tabaci. These genes were composed of three representative genes related to SA pathway (*PAD4*, *EDS1*, and *NPR1*) and one gene related to JA pathway (*PDF1.2*). After 24 hpi with either pathogen in Col-0, the SA marker genes, *PAD4* and *EDS1*, and JA marker gene *PDF1.2*were strongly induced, but the level of induction of these genes was significantly lower in the *Atfdh1* mutant against both host

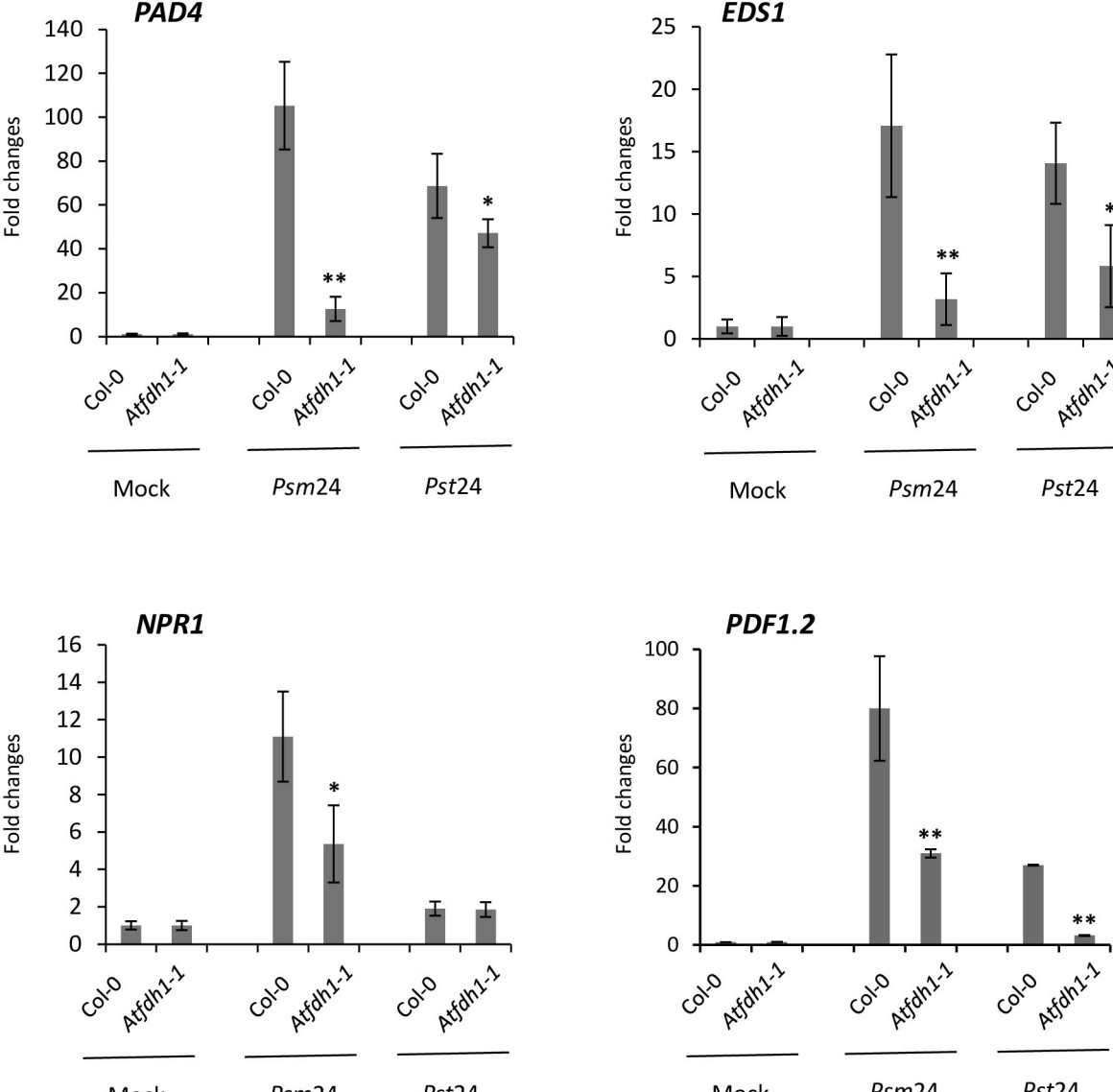

**Fig 3. Patterns of gene expression associated with SA-mediated defense signaling pathways in wild type (Col-0) and *Atfdh1* mutant (*Atfdh1-1*).** The expression of SA-mediated defense-related genes were examined after 24 hrs post inoculation (hpi) in response to host, *P. syringae* pv. maculicola, and nonhost pathogen, *P. syringae* pv. tabaci. Four weeks old seedlings were flood-inoculated with the concentration of $1 \times 10^5$ CFU/ml bacterial suspension. Each column is the fold change of gene expression as determined by RT-qPCR at 24 hpi in pathogen-inoculated samples. The relative gene expression values normalized by *Ubiquitin5* (*UBQ5*) and *Elongation factor 1 alpha* (*EF1α*) are represented as n-fold compared to the mock-treated plants. Fold changes are over the non-treated Col-0 or mutants. Asterisks above bars represent statistically significant differences in comparison with wild-type using Student's *t*-test (P < 0.05). The gene expression was examined with four biological samples (three technical repeats for each sample). *Psm24*: 24 hours after the inoculation of *P. syringae* pv. maculicola, *Pst24*: 24 hours after inoculation of *P. syringae* pv. tabaci.

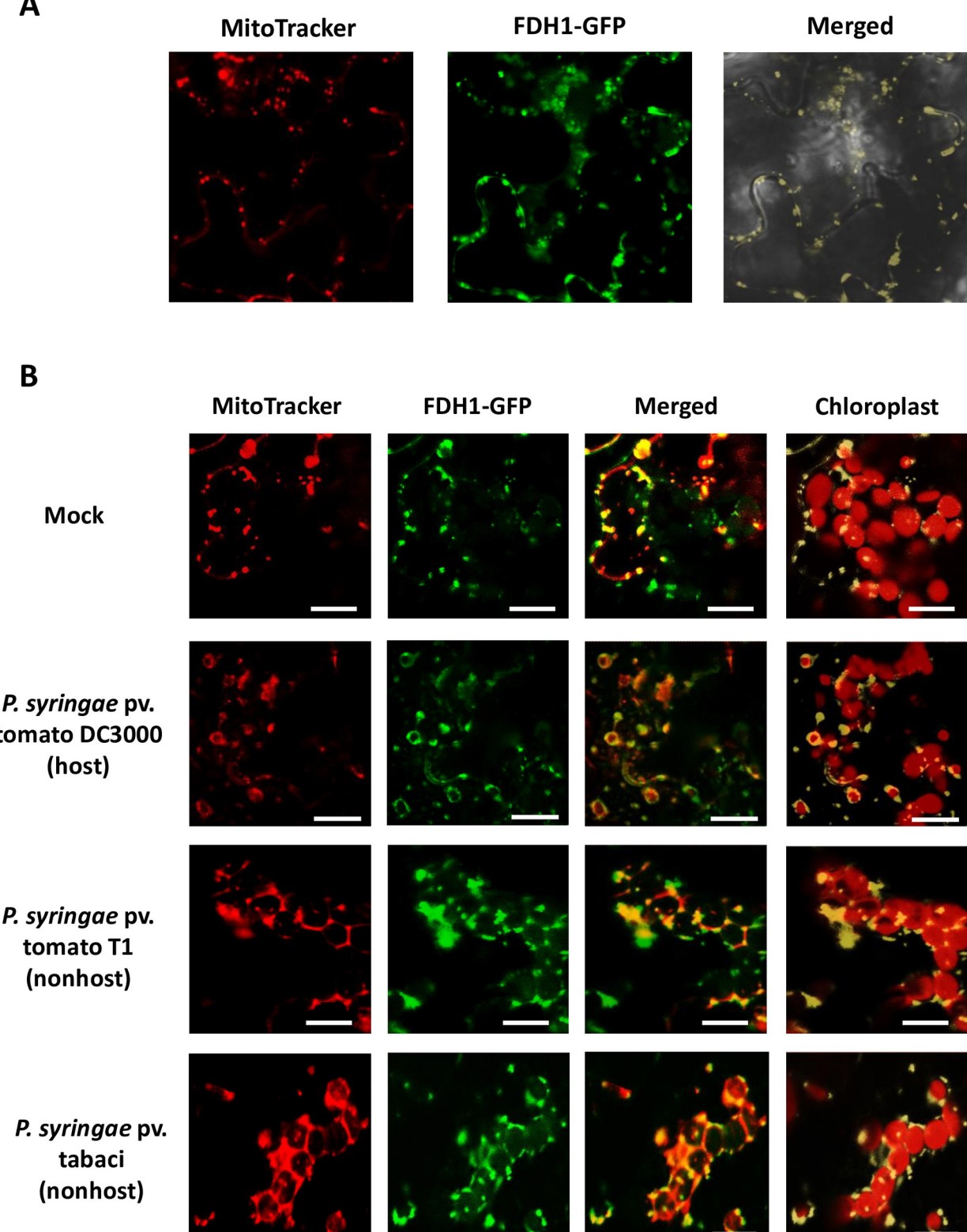

**Fig 4. Localization of AtFDH1 in Arabidopsis leaves in response to host and nonhost bacterial pathogens.** AtFDH1 is localized to mitochondria (A). The expression and localization of AtFDH1-GFP was observed in peeled adaxial epidermal cells from leaves of transgenic Arabidopsis lines expressing *AtFDH1-GFP*. The protein localization was also examined in detached leaf samples after the treatment of *P. syringae*

pv. tomato DC3000 ($1\times10^5$ CFU/ml), *P. syringae* pv. tomato T1 ($1\times10^5$ CFU/ml), and *P. syringae* pv. tabaci ($1\times10^5$ CFU/ml) under a confocal microscope (B). MitoTracker red dye was used to stain mitochondria. Bars = 10 µm. For MitoTracker Red, a 561 nm excitation, 570–620 nm emission filter was used. Red channel (680 nm emission filter) shows chlorophyll auto fluorescence in chloroplasts of mesophyll cells. Green channel shows the fluorescence signal of AtFDH1-GFP in mitochondria and outer membrane of chloroplast. In merged images, the fluorescence signal overlapped by MitoTracker and AtFDH1-GFP is shown in yellow. In chloroplast image, yellow represents a merged signal of FDH1-GFP localization in mitochondria and chloroplast.

and nonhost pathogens, compared to Col-0 (Fig 3). *NPR1* was significantly induced at 24 hpi with the host pathogen in wild-type Col-0 and decreased 5-fold in the *Atfdh1* mutant. *NPR1* was not significantly induced after inoculation with the nonhost pathogen in both mutant and wild-type lines. These results suggest that *AtFDH1* plays a role in plant defense responses via SA and JA mediated defense pathways.

## AtFDH1 localizes predominantly in mitochondria and targets to chloroplasts for bacterial defense responses

Localization of FDH1 in mitochondria and/or chloroplast has been the subject of extensive debate [23, 27–30]. We cloned *AtFDH1* to be expressed under its native promoter and fused it to the C-terminal of *Green Fluorescent Protein* (*GFP*) gene and transiently expressed it in *N. benthamiana*. The results showed that AtFDH1-GFP predominantly localizes to mitochondria (S4 Fig). We generated Arabidopsis stable lines expressing *AtFDH1-GFP* in Col-0, and the localization of AtFDH1-GFP in mitochondria was confirmed using the live cell mitochondrial stain MitoTracker (Fig 4A). Upon challenging the plant with host (*P. syringae* pv. tomato DC3000) and nonhost (*P. syringae* pv. tomato T1) bacterial pathogens, AtFDH1-GFP signal was also found at the outer envelope membrane of chloroplasts in addition to mitochondria (Fig 4B). Similar results were observed with another nonhost pathogen (*P. syrinage* pv. phaseolicola) (S5 Fig). The expression of AtFDH1-GFP (observed as green fluorescence) was remarkably higher after nonhost (*P. syringae* pv. tomato T1 and *P. syringae* pv. phaseolicoa) and host (*P. syringae* pv. tomato DC3000) pathogen treatments than the expression in the detached leaf sample without pathogen challenge (Figs 4B and S5).

Using the light-sheet microscope, the movement and co-localization of mitochondria with chloroplasts were observed after the inoculation of nonhost pathogen, *P. syringae* pv. tomato T1 and *P. syringae* pv. tabaci. By performing a time lapse image of FDH1 localization in the transgenic Arabidopsis line expressing *FDH1-GFP*, we found that mitochondrial specific FDH1 localization was highly motile after inoculation with a nonhost pathogen. As shown in the Fig 5, the majority of mitochondria localized FDH1 was in the vicinity of chloroplasts in outer-membrane regions. The arrow in each image (every 1.5 min) shows the movement of mitochondria localized FDH1 around chloroplasts. In the time-lapse image, we observed that the mitochondria (FDH1::GFP) aggregate with chloroplasts and later goes apart from the chloroplast. After this event, other mitochondria localized FDH1 translocate again to chloroplasts, and we observed the event of co-localization continuously during the 15 min of time-lapse imaging (S1 Video). These results suggest that in response to nonhost pathogen, FDH1 co-localizes to both mitochondria and chloroplast and the localization of FDH1 in chloroplast is transient.

To further investigate the specific localization of AtFDH1 in mitochondria and chloroplast upon host and nonhost pathogens, the protein of mitochondria and chloroplast were isolated separately from AtFDH1-GFP expressing plants and examined for the presence of AtFDH1 protein. Immunoblot analysis revealed that in total protein extract, AtFDH1-GFP accumulates in response to host and nonhost pathogens at 2 and 4 hpi, which coincides with the result of RT-qPCR (Figs 6 and S3). To validate the localization of AtFDH1, we isolated mitochondria

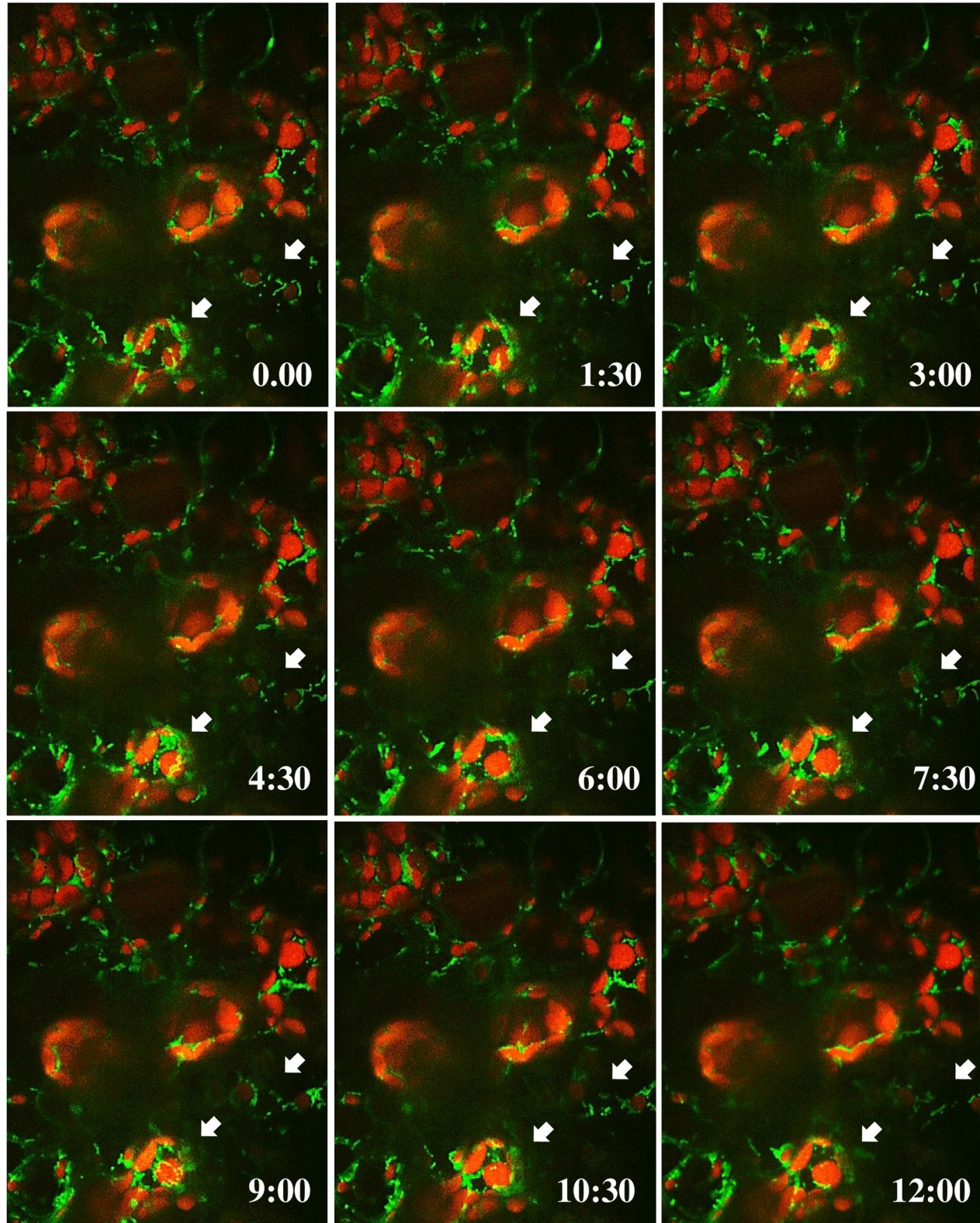

**Fig 5. Time lapse imaging of co-localization of AtFDH1 in mitochondria and chloroplasts in response to nonhost pathogen.** Detached leaves from transgenic Arabidopsis plants expressing *AtFDH1-GFP* driven by *AtFDH1* promoter were treated with *P. syrinage* pv. tomato T1 ($1\times10^5$ CFU/ml). *AtFDH1* expression was monitored 40 min after the pathogen infection by using a light-sheet fluorescence microscope (Carl Zeiss, Germany).

The images were observed for 15 min by time lapse imaging at 30 second intervals in Z-stack mode. For every 30 sec, the AtFDH1-GFP and chlorophyll fluorescence of chloroplast was captured and both live streaming videos were merged to generate time lapse video simultaneously. Images were taken from detached leaf samples 60 min after treatment of *P. syrinage* pv. tomato T1. Times shown at bottom-right of each image are in minutes:seconds.

and chloroplasts from AtFDH1-GFP expressing plants upon inoculation with host or nonhost pathogen. Mitochondria and chloroplast proteins were individually extracted and subjected to immunoblot analyses. AtFDH1-GFP protein was detected in mitochondria prior to pathogen infection, and the protein amount increased significantly after host or nonhost pathogen infection (Fig 6). By contrast, AtFDH1-GFP protein was not detected in the chloroplast protein extract prior to pathogen infection. Consistent with the cell biology data, AtFDH1-GFP was detected in the chloroplast protein extract after infection with host or nonhost pathogen infection (Fig 6). However, the accumulation of AtFDH1 protein was only found in the chloroplast protein fraction at 2 hours after infection with nonhost pathogen, and the expression level was much higher at 4 hpi when compared to host pathogen (Fig 6).

## Discussion

FDH enzyme is found in various organisms such as bacteria, yeast, and plants. This protein has been reported to function during various abiotic and biotic stress responses. Expression of *FDH* is strongly induced during various abiotic and biotic stress responses such as pathogen, hypoxia, chilling, drought, dark, wounding and iron deficiency [21, 23, 24]. There is only one study showing that FDH1 is involved in regulating plant cell death and defense responses against bacterial pathogens in pepper plants [23]. In this study, mitochondrial targeting of FDH1 played an important role in PCD- and SA-dependent defense response, and silencing of *FDH1* attenuates resistance against *X. campestris* pv. vesicatoria pathogen in pepper plants. Our study demonstrates that *FDH1* is involved in plant innate immunity against both host and nonhost bacterial pathogens. Nonhost disease resistance is the most common form of plant defense against various pathogens [2, 5, 52–54]. HR cell death are typical symptoms in response to ETI-triggered nonhost resistance in plants [20]. ROS produced in various cellular compartments, including chloroplasts, mitochondria, and peroxisomes have been proposed to act as signals for HR and PCD [55–57]. Chloroplasts are the main source of ROS during various environmental stresses, including plant-pathogen interactions [33, 36]. In addition, ROS generated in mitochondria (mtROS) has been described in several studies to be an important factor in inducing HR cell death against plant pathogens [38, 57]. Possibly both chloroplasts and mitochondria have a role in nonhost resistance against invading bacterial pathogens. In this study, we demonstrate that the protein encoded by a single *FDH1* gene in the nuclear genome is targeted to both mitochondria and chloroplasts in response to wounding and bacterial pathogens. Chloroplast localization of FDH1 was more abundant after inoculation with nonhost pathogens (Figs 4 and 5), thus suggesting a probable role of chloroplasts in nonhost disease resistance. A previous study has shown that chloroplast generated ROS is required for nonhost disease resistance in Arabidopsis [58]. In addition to nonhost resistance, we also show that *FDH1* plays a role in basal and gene-for-gene resistance in Arabidopsis. It is intriguing that the silencing of *NbFDH1* did not compromise basal resistance in *N. benthamiana*. Since the silencing of *NbFDH1* decreased *NbFDH1* transcripts by ~70%, we speculate that this might not be sufficient to compromise basal resistance. By contrast, the complete knockout of *AtFDH1* in Arabidopsis compromised basal resistance.

Our study identified a dual-targeting role for AtFDH1 during plant defense responses against bacterial pathogens. Dual targeting of FDH1 to mitochondria and chloroplasts may

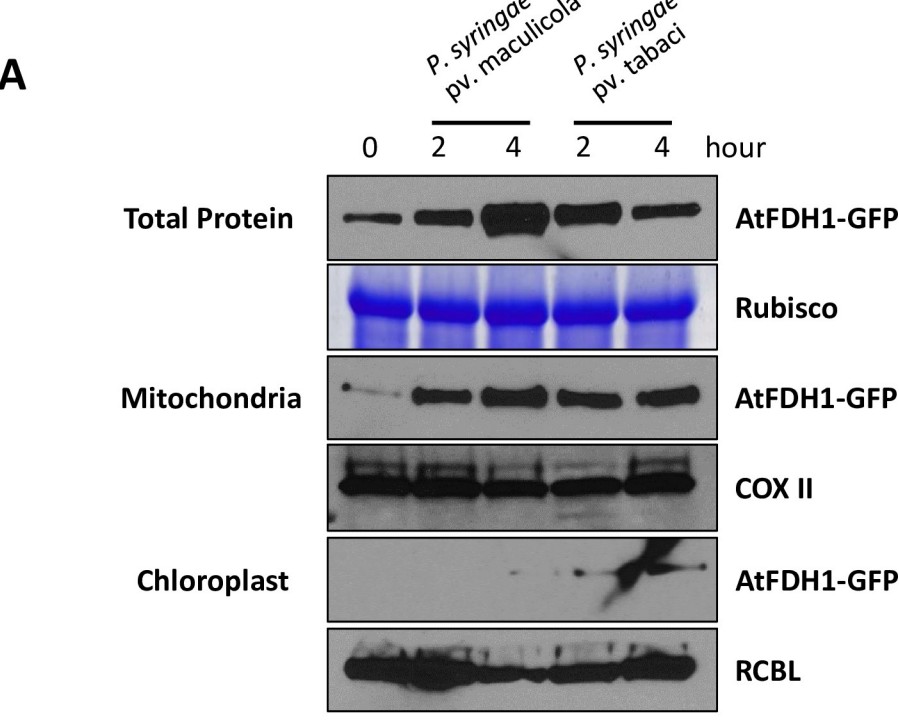

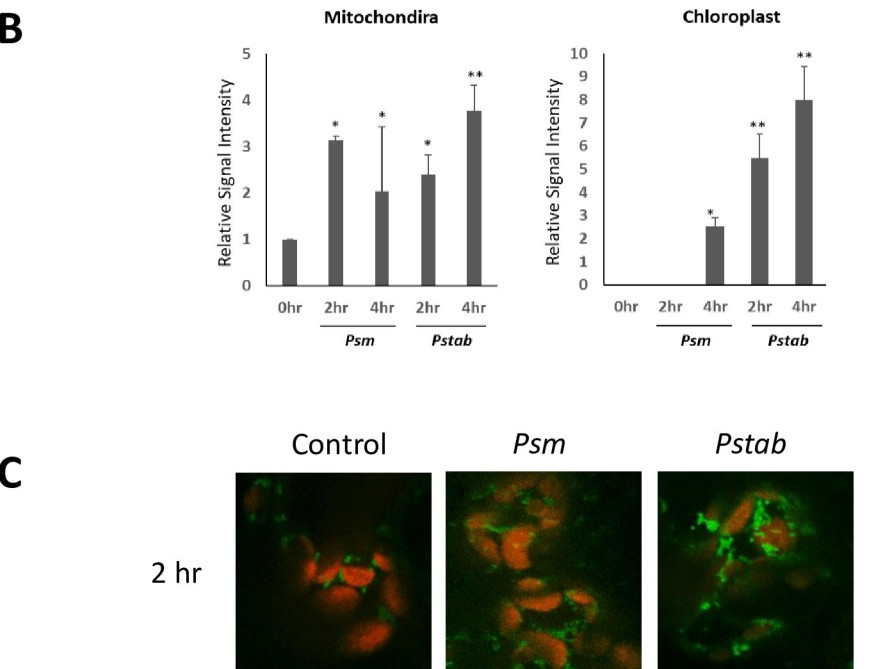

**Fig 6. Accumulation of AtFDH1 protein in response to host or nonhost pathogens in mitochondria and chloroplasts.** The 8-week-old Arabidopsis wild-type (Col-0) were flood-inoculated with the concentration of $1 \times 10^5$ CFU/ml bacterial suspension of *P. syringae* pv. maculicola (host) or *P. syringae* pv. tabaci (nonhost) pathogens. Leaf samples were collected at 0, 2, and 4 hpi for the protein extraction, and 3 µg protein from mitochondria or chloroplast was used for the immunoblot assay (A). Because no AtFDH1-GFP was visible in chloroplast samples with 3 µg total protein, a total of 28 µg was used. Rubisco: internal control for total protein (bromophenol blue stained gel), COXII: mitochondria marker protein detected using polyclonal COXII antisera (Agrisera), RBCL: chloroplast marker protein

detected using polylonal Rbcl antisera (Abiocod). The band intensities on western blot membrane were measured using ImageJ software (https://imagej.nih.gov/ij/) for comparison (B). The bars show the mean values and standard deviations of the mean (* $0.05 > P$ and ** $0.01 > P$). The experiments were replicated three times. The level of AtFDH1-GFP was determined under confocal microscope after infection of host (*P. syringae* pv. maculicola) and nonhost pathogen (*P. syringae* pv. tabaci) (C).

be necessary for effective signaling during plant defense against bacterial pathogens. In the Arabidopsis nuclear genome, approximately 20–25% of the genes encode proteins that are targeted to either mitochondria or chloroplasts [59]. It has been reported that some proteins target to both mitochondria and chloroplast, and might be more common than thought. However, their functions are not well understood, especially for plant disease resistance [59–62]. FDH1 has a putative mitochondrial signal peptide, although AtFDH1 has been reported to localize to either mitochondria or chloroplasts [30, 63–65]. Therefore, FDH1 localization in plants remains controversial. One study showed the dual localization of AtFDH1 in both chloroplasts and mitochondria when AtFDH1 is overexpressed in transgenic Arabidopsis and tobacco plants [26, 66]. It is also reported that the N-terminal region of AtFDH1 is predicted to contain the signal peptide region that could target it to chloroplasts as well as mitochondria [67]. This N-terminal sequence of AtFDH1 is quite different from potato, barley, and rice, suggesting AtFDH1 localizing in chloroplast could occur under certain conditions [65]. In our study, the localization of AtFDH1 in chloroplast was mainly detected under the conditions of wounding and pathogen stresses (Figs 4 and 5). As previously described, *FDH1* is highly induced under various stress conditions [65]. It is possible that the localization of FDH1 in chloroplast is low and transient to be detected under non-stress conditions, and this could cause controversy of the FDH1 localization in mitochondria or chloroplasts or both.

There are few reports that suggest FDH1 may have a role in biotic stress response in plants. As mentioned above, FDH1 has been shown to play a role in disease resistance in pepper against a bacterial pathogen [23]. FDH1 and Calreticulin-3 precursor (CRT3) directly interacts with the helicase domain of *Cucumber mosaic virus* (CMV) isolate-P1, suggesting that FDH1 has an important role in plant disease resistance [68]. CRT3 is localized in the endoplasmic reticulum (ER) lumen, and has been known to associate with abiotic stress response and plant immunity [69–71]. FDH1 directly interacts with RING-type ubiquitin ligase Keep on Going (KEG), which is localized in trans-golgi and early endosomes [25]. In Arabidopsis, the loss of function in KEG disrupts the secretion of the apoplastic defense proteins such as pathogenesis-related PR1, which indicates the involvement of KEG in plant immunity [72]. There are several reports describing the ROS-based signal transmission between mitochondria and chloroplasts [33, 73–75]. Possibly, FDH1 protein could be associated with a signal transduction pathway for the production of chloroplast-derived ROS.

## Conclusions

In this study, we demonstrated the possible role of chloroplast-dependent pathway that regulates plant innate immunity, probably through mitochondria-to-chloroplast integrated ROS signaling. Even though mitochondria are the main source of ROS, chloroplasts also play a role in producing ROS during stress responses in plants. However, the signal transduction between these organelles for coordinated production of ROS is not well understood. Characterization of molecular functions of FDH1-interactors in both mitochondria and chloroplasts would provide insight into the role of FDH1 in cross-talk between these organelles during biotic and abiotic stress responses.

## Supporting information

**S1 Fig. Sequence alignment of FDH1 protein from *N. benthamiana* (NbFDH1), tobacco (NtFDH1), tomato (SlFDH1), and Arabidopsis (AtFDH1).** Sequence information was obtained from the public database; TAIR, NCBI GenBank, and Sol Genomics Network. The software MEGA-X [76] was used for sequence alignment. The amino acid colors were in accordance with the default coloring schemes of ClustalX alignment, which depends on both residue type and the pattern of conservation within a column (http://www.clustal.org/clustal2/).
(PPTX)

**S2 Fig. The expression of the *NbFDH1* gene is reduced in *NbFDH1*-silenced *N. benthamiana* plants.** Two weeks old *N. benthamiana* seedlings were inoculated with TRV1 + TRV::00 (control) or TRV1 + TRV::*NbFDH1*. Three weeks after TRV inoculation, leaf samples from three different biological replicates for each construct were collected, and gene expression was measured by RT-qPCR. *NbActin* was used as internal control for normalization. Bars represent mean, and error bars represent standard deviation for three biological replicates (four technical replicates for each biological sample). Asterisk represents statistical significance that was determined using Student's *t*-test, (P < 0.01).
(PPTX)

**S3 Fig. *AtFDH1* is upregulated upon inoculation with host and nonhost pathogens in wild-type Col-0, and some defense-related genes are differentially expressed in *Atfdh1* mutant.** (A) Gene expression patterns of *AtFDH1* against *P. syringae* bacterial pathogen in Arabidopsis. This data was obtained from Arabidopsis eFP Browser at bar.utoronto.ca [77]. (B) *AtFDH1* is induced by host and nonhost pathogen inoculations. Four-weeks-old Arabidopsis wild-type (Col-0) were flood-inoculated with host (*P. syringae* pv. maculicola, *Psm*) or nonhost (*P. syringae* pv. tabaci, *Pstab*) pathogens. The 24 hours after inoculation, leaves were harvested, total RNA was extracted, and subject to RT-qPCR using *AtFDH1* specific primers. *AtActin* was used as an internal control for normalization. Bars represent mean, and error bars represent standard deviation for three biological replicates (four technical replications for each biological replicate). Asterisks represent statistical significance as determined using Student's *t*-test, (P < 0.01).
(PPTX)

**S4 Fig. Localization of AtFDH1 in *N. benthamiana*.** For *Agrobacterium*-mediated transient assay, a binary vector containing *GFP* gene fused to the C-terminal of *AtFDH1* was transformed into the *A. tumefaciens* strain GV3101. The *Agrobacterium* suspension was ($5 \times 10^7$ CFU/ml) was infiltrated using a needleless syringe into *N. benthamiana* leaves, and the green fluorescence representing AtFDH1 localization was observed 3 days after the agroinfiltration. Red channel (a 561 nm excitation, 570–620 nm emission filter) shows mitochondria stained with MitoTracker dye and green channel shows AtFDH1-GFP localization. Bars = 10 μm.
(PPTX)

**S5 Fig. Localization of AtFDH1 in Arabidopsis leaves.** The expression and localization of AtFDH1-GFP was observed in detached (no stress) and peeled adaxial epidermal cells (pathogen stress) from leaves of transgenic Arabidopsis lines expressing *AtFDH1-GFP* in Col-0. The protein localization was also examined in detached leaf samples 1-hr after the treatment of *P. syringae* pv. tomato DC3000 ($1 \times 10^5$ CFU/ml) and *P. syringae* pv. phaseolicola ($1 \times 10^5$ CFU/ml). Red channel (a 561 nm excitation, 570–620 nm emission filter), showing chloroplast; green channel showing AtFDH1-GFP. Bars = 10 μm.
(PPTX)

**S6 Fig. Western blot analysis for the expression of AtFDH1-GFP in mitochondria and chloroplast after infection of host (*P. syringae* pv. *maculicola*) and nonhost (*P. syringae* pv. *tabaci*) pathogens.** (A) AtFDH1-GFP expression in total protein, (B) Internal control for rubisco expression for total protein, (C) AtFDH1-GFP expression in mitochondrial protein, (D) COXII expression for the internal control of mitochondrial protein, (E) AtFDH1-GFP expression in chloroplast protein, (F) RBCL expression for the internal control of chloroplast. (PPTX)

**S1 Video. Live video for time lapse imaging of co-localization of AtFDH1 in mitochondria and chloroplasts in response to nonhost pathogen.** Experimental methods are described in Fig 5.
(AVI)

**S1 Raw images.**
(PDF)

**S1 Data.**
(XLSX)

**S2 Data.**
(XLSX)

**S3 Data.**
(XLSX)

## Acknowledgments

We thank Dr. Jin Nakashima for his assistance with cellular imaging and Dr. Elison Blancaflor for his critical reading of the manuscript. The authors also thank all technical support and research assistance provided from Dr. Mysore's Lab for this study.

## Author Contributions

**Funding acquisition:** Kirankumar S. Mysore.

**Investigation:** Seonghee Lee.

**Methodology:** Seonghee Lee, Ramu S. Vemanna, Sunhee Oh, Clemencia M. Rojas, Amita Kaundal, Taegun Kwon, Hee-Kyung Lee, Muthappa Senthil-Kumar.

**Supervision:** Seonghee Lee, Kirankumar S. Mysore.

**Writing – original draft:** Seonghee Lee, Ramu S. Vemanna, Clemencia M. Rojas, Youngjae Oh, Kirankumar S. Mysore.

**Writing – review & editing:** Seonghee Lee, Clemencia M. Rojas, Youngjae Oh, Amita Kaundal, Taegun Kwon, Muthappa Senthil-Kumar, Kirankumar S. Mysore.

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
