## [Decision Letter · Decision Letter 0]

23 Feb 2021

PONE-D-20-40925

Formate Dehydrogenase (FDH1) Localizes to Both Mitochondria and Chloroplast to Play a Role in Host and Nonhost Disease Resistance

PLOS ONE

Dear Dr. Lee,

Thank you for submitting your manuscript to PLOS ONE. After careful consideration, we feel that it has merit but does not fully meet PLOS ONE’s publication criteria as it currently stands. Therefore, we invite you to submit a revised version of the manuscript that addresses the points raised during the review process.

This was an interesting topic that was generally well-received by the reviewers. However, in addition to several minor concerns that both reviewers raised, and that can likely be easily addressed, Reviewer 2 indicated that the localization studies were not yet compelling. I am in agreement that the use of a co-reporter will be useful here. If a suitable reporter is unavailable, other means of improving understanding of FDH1 localization would also be acceptable. In addition, comments made regarding the western blot analysis that should be addressed.

We look forward to receiving your revised manuscript.

Kind regards,

Richard A Wilson

Academic Editor

PLOS ONE

Journal Requirements:

2.PLOS ONE now requires that authors provide the original uncropped and unadjusted images underlying all blot or gel results reported in a submission’s figures or Supporting Information files. This policy and the journal’s other requirements for blot/gel reporting and figure preparation are described in detail at https://journals.plos.org/plosone/s/figures#loc-blot-and-gel-reporting-requirements and https://journals.plos.org/plosone/s/figures#loc-preparing-figures-from-image-files. When you submit your revised manuscript, please ensure that your figures adhere fully to these guidelines and provide the original underlying images for all blot or gel data reported in your submission. See the following link for instructions on providing the original image data: https://journals.plos.org/plosone/s/figures#loc-original-images-for-blots-and-gels.

3.Thank you for stating the following in the Financial Disclosure section:

We note that one or more of the authors are employed by a commercial company: Noble Research Institute, LLC

Reviewers' comments:

Reviewer's Responses to Questions

**Comments to the Author**

1. Is the manuscript technically sound, and do the data support the conclusions?

Reviewer #1: Yes

Reviewer #2: Partly

2. Has the statistical analysis been performed appropriately and rigorously? 

Reviewer #1: Yes

Reviewer #2: No

3. Have the authors made all data underlying the findings in their manuscript fully available?

Reviewer #1: Yes

Reviewer #2: Yes

4. Is the manuscript presented in an intelligible fashion and written in standard English?

Reviewer #1: Yes

Reviewer #2: No

5. Review Comments to the Author

Reviewer #1: The manuscript is overall well described the goals, importance and results; properly designed and soundly carried out, emphasizing the important of FDH1 in plant defense responses.

Minor points are;

In abstract, authors argued that SA signaling is involved in FDH1-mediated defense response. However, Fig. 3 and a corresponding chapter showed the role of both SA and JA in FDH1 activity. Authors should revise the abstract according to the Fig. 3 results, and may discuss potential implications on SA and JA coordination (?) or common understanding of SA/JA antagonisms (if possible, additional JA-marker genes could be added, and/or timely compared with SA-marker genes).

Line 245 and 398; the efficacy of NbFDH1-silencing was differently described between line 245 (70%) and line 398 (50%). Those ~20% may have somewhat critical effects on physiology, e.g., may explain different results observed between tomato-silencing and Arabidopsis-KO lines on the infection of host pathogens (Fig. 1C vs. 2B).

Line 292: chapter “AtFDH1 is … pathogens” compared the upregulation of FDH1 expressions in response to host or nonhost bacteria, and proposed a greater role of FDH1 in nonhost disease resistance. Is it possible that induction/response of FDH1 upregulation by nonhost bacteria is slower (or faster) than by host pathogens? RT-PCR on additional time points could confirm/strengthen the authors’ conclusion.

Reviewer #2: The manuscript entitled “Formate Dehydrogenase (FDH1) Localizes to Both Mitochondria and Chloroplast to Play a Role in Host and Nonhost Disease Resistance” by Lee et al. described isolation of a FDH1 gene encoding a formate dehydrogenase from two plant species, Nicotiana benthamiana and Arabidopsis thaliana. The authors concluded that the gene provides nonhost, basal and effector triggered immunity. Furthermore, they concluded that FDH1 in Arabidopsis is localized to mitochondria and following bacterial infection it moves to chloroplast.

It is an interesting study reporting new knowledge in the area of plant immunity.

Major Comments:

1. Localization section requires additional studies. It is hard to follow the photos relevant to mitochondrial and chloroplastic localization of FDH1. It will be best to co-localize FDH1 with a reporter protein; i. e, an RFP fused protein that has been already established to be a mitochondrial or chloroplast protein. Close-up photos will also help the readers to follow the changes in accumulation patterns following infection.

2. The western-blot looks good; but quantification has not made to claim that there is an enhanced accumulation of the fusion protein with time following infection (Figure 6).

This is a very interesting preliminary observation that FDH1 starts to accumulate in chloroplast following infection. Is not it possible that the protein is directly targeted to chloroplast from cytoplasm following infection? Note that in lane 3, the total protein is much more than that in mitochondria suggesting an increased synthesis of FDH1 following infection.

Minor Comments:

1. It is not evident in this study if FDH1 localization to mitochondria or chloroplast is involved in host and nonhost disease resistance; therefore, title may be revised.

2. The basal resistance section is less developed. How was it measured?

3. Lines 60 - 61: “The efficacy of nonhost disease resistance is based on the recognition of pathogen-associated molecular patterns (PAMPs) and/or pathogen effectors.” This statement should be revised as not all nonhost resistance mechanisms are controlled by receptors for PAMPs. A review of the nonhost resistance mechanisms/genes should be included in “Introduction.”

4. Line 252: The sentence starting with “In correlation with ….” should be revised.

5. Line 293: May delete the first section of the sentence up to the comma.

6. Lines 307-308: The sentence should be deleted. Basal resistance is not even referred in the previous paragraph.

7. Lines 324 -325: “AtFDH1 localizes predominantly in mitochondria, but translocates to chloroplasts in response to abiotic and biotic stresses” – No evidence is provided in support of the translocation of the protein from mitochondria to chloroplast.

8. The manuscript would be benefited from editing for English.

6. PLOS authors have the option to publish the peer review history of their article (what does this mean?). If published, this will include your full peer review and any attached files.

Reviewer #1: No

Reviewer #2: No

---

## [Author Response · Author response to Decision Letter 0]

14 Dec 2021

Dear Editor, 

Thank you for the valuable comments on our manuscript. We have followed the reviewers’ comments and thoughtfully revised the manuscript. 

Please see below our responses and let us know for any questions. 

Best Regards, 

Seonghee Lee

Dear Dr. Lee,

Thank you for submitting your manuscript to PLOS ONE. After careful consideration, we feel that it has merit but does not fully meet PLOS ONE’s publication criteria as it currently stands. Therefore, we invite you to submit a revised version of the manuscript that addresses the points raised during the review process.

This was an interesting topic that was generally well-received by the reviewers. However, in addition to several minor concerns that both reviewers raised, and that can likely be easily addressed, Reviewer 2 indicated that the localization studies were not yet compelling. I am in agreement that the use of a co-reporter will be useful here. If a suitable reporter is unavailable, other means of improving understanding of FDH1 localization would also be acceptable. In addition, comments made regarding the western blot analysis that should be addressed.

Reviewer #1: The manuscript is overall well described the goals, importance and results; properly designed and soundly carried out, emphasizing the important of FDH1 in plant defense responses.

Minor points are;

In abstract, authors argued that SA signaling is involved in FDH1-mediated defense response. However, Fig. 3 and a corresponding chapter showed the role of both SA and JA in FDH1 activity. Authors should revise the abstract according to the Fig. 3 results, and may discuss potential implications on SA and JA coordination (?) or common understanding of SA/JA antagonisms (if possible, additional JA-marker genes could be added, and/or timely compared with SA-marker genes).

-> That is correct. We found the important role of both SA and JA for host and nonhost resistance against bacterial pathogens. The statement was corrected. 

Line 245 and 398; the efficacy of NbFDH1-silencing was differently described between line 245 (70%) and line 398 (50%). Those ~20% may have somewhat critical effects on physiology, e.g., may explain different results observed between tomato-silencing and Arabidopsis-KO lines on the infection of host pathogens (Fig. 1C vs. 2B).

-> As shown in the supplementary Figure 2, the silencing of NbFDH1 is approximately 70%. We corrected the statement in line 402. 

Line 292: chapter “AtFDH1 is … pathogens” compared the upregulation of FDH1 expressions in response to host or nonhost bacteria, and proposed a greater role of FDH1 in nonhost disease resistance. Is it possible that induction/response of FDH1 upregulation by nonhost bacteria is slower (or faster) than by host pathogens? RT-PCR on additional time points could confirm/strengthen the authors’ conclusion.

-> As shown in supplementary Figure 3 and 5, we found that expression level of FDH1 was much higher in response to nonhost pathogen than by host pathogen. In qRT-PCR assay, we only determined the FDH1 expression at 24hpi; thus, it would be difficult to explain if the gene expression would be faster or slower in response host vs. nonhost pathogens. However, in Figure 6, FDH1 accumulation was elevated faster in response to nonhost pathogen (Pst, 2hpi) at chloroplast sample. The statement was revised in line 370. 

Reviewer #2: The manuscript entitled “Formate Dehydrogenase (FDH1) Localizes to Both Mitochondria and Chloroplast to Play a Role in Host and Nonhost Disease Resistance” by Lee et al. described isolation of a FDH1 gene encoding a formate dehydrogenase from two plant species, Nicotiana benthamiana and Arabidopsis thaliana. The authors concluded that the gene provides nonhost, basal and effector triggered immunity. Furthermore, they concluded that FDH1 in Arabidopsis is localized to mitochondria and following bacterial infection it moves to chloroplast.

It is an interesting study reporting new knowledge in the area of plant immunity.

Major Comments:

1. Localization section requires additional studies. It is hard to follow the photos relevant to mitochondrial and chloroplastic localization of FDH1. It will be best to co-localize FDH1 with a reporter protein; i. e, an RFP fused protein that has been already established to be a mitochondrial or chloroplast protein. Close-up photos will also help the readers to follow the changes in accumulation patterns following infection.

-> We have used MitoTracker red dye to determine the subcellular localization of FDH1, because previous studies suggested that FDH1 localized in mitochondria and possibly cloroplast. MitoTracker was also used as mitochondrial staining dye and validated the localization of FDH1 in pepper (Choi et al., 2014, Plant Physiology). In addition, there are a number of studies published using MitoTracker as the dye of mitochondria straining (few examples - labeling mitochondria with MitoTracker dyes, Cold Spring Harbor Protocols, 2010; Sorvina et al., 2018, “Mitochondrial imaging…. “, Scientific reports; Jeena et al., 2017, “Mitochondrial localization….”, Nature communication). We agree our localization image was not clearly zoomed in, so another high quality of close-up image (Figure 4) was included to show the localization of FDH1 in mitochondria. 

Also, in Figure 5, we did not use RFP fused protein for examining the chloroplast localization. Instead of using RFP fusing protein, we confirmed chloroplast with red autofluorescence of chlorophyll in mesophyll cells. Because it is well known that chloroplasts exhibit very strong autofluorescence in red, with a peak of approximately 680 nm. Thus, we used red channel to show chlorophyll auto fluorescence in chloroplasts of the epidermal cell region (this information was added in line 516). There are numerous studies to use the red autofluorescence of chlorophyll to determine chloroplast. We agree that the image resolution of Figure 5 was not in good quality to understand colocalization of FDH1 in mitochondria and chloroplast. This Figure has been revised for better explanation of the distributional change of FDH1-GFP in response to bacterial pathogens in mitochondria and chloroplast. Please see the revised Figure 4. 

2. The western-blot looks good; but quantification has not made to claim that there is an enhanced accumulation of the fusion protein with time following infection (Figure 6).

-> That is correct, so we measured western blot band intensities. Figure 6 was revised. 

This is a very interesting preliminary observation that FDH1 starts to accumulate in chloroplast following infection. Is not it possible that the protein is directly targeted to chloroplast from cytoplasm following infection? Note that in lane 3, the total protein is much more than that in mitochondria suggesting an increased synthesis of FDH1 following infection.

-> There would be a possibility that FDH1 could be targeted to chloroplast from cytoplasm. Based on our confocal microscope observation in many different experiments, FDH1 is mainly localized in mitochondria, and the expression is drastically increased after nonhost bacterial pathogens. We also see the movement of FDH1 expressing mitochondria is much faster and colocalized to chloroplast after nonhost pathogen infection than host pathogen and water control. Figure 6 is updated and showed the increased synthesis of FDH1 in response to nonhost pathogen (also please see the supplementary Figure 5). 

Minor Comments:

1. It is not evident in this study if FDH1 localization to mitochondria or chloroplast is involved in host and nonhost disease resistance; therefore, title may be revised.

-> We agrees the comment, and title has been revised. 

2. The basal resistance section is less developed. How was it measured?

-> In this study, we found that FDH1 is upregulated against both host and nonhost pathogens. Also as shown in Figure 2, fdh1 mutant plants were more susceptible to host pathogen, and the bacterial population in infected leaf tissues were much higher than the bacterial population in wide type. Also similar result showed in another group study in pepper (Choi et al., 2014). 

3. Lines 60 - 61: “The efficacy of nonhost disease resistance is based on the recognition of pathogen-associated molecular patterns (PAMPs) and/or pathogen effectors.” This statement should be revised as not all nonhost resistance mechanisms are controlled by receptors for PAMPs. A review of the nonhost resistance mechanisms/genes should be included in “Introduction.”

-> This sentence was revised as shown in line 63-70. 

4. Line 252: The sentence starting with “In correlation with ….” should be revised.

-> This sentence was correct. 

5. Line 293: May delete the first section of the sentence up to the comma.

-> Corrected as shown in line 280. 

6. Lines 307-308: The sentence should be deleted. Basal resistance is not even referred in the previous paragraph.

-> corrected (line 317).

7. Lines 324 -325: “AtFDH1 localizes predominantly in mitochondria, but translocates to chloroplasts in response to abiotic and biotic stresses” – No evidence is provided in support of the translocation of the protein from mitochondria to chloroplast.

-> Yes we agree that, and the sentence was changed. 

8. The manuscript would be benefited from editing for English.

- > The manuscript has been thoughtfully read and revised for any potential grammatical errors in English.

---

## [Decision Letter · Decision Letter 1]

4 Jan 2022

PONE-D-20-40925R1Functional Role of Formate Dehydrogenase (FDH1) for Host and Nonhost Disease Resistance Against Bacterial PathogensPLOS ONE

Dear Dr. Lee,

Thank you for submitting your manuscript to PLOS ONE. After careful consideration, we feel that it has merit but does not fully meet PLOS ONE’s publication criteria as it currently stands. Therefore, we invite you to submit a revised version of the manuscript that addresses the points raised during the review process.

The manuscript is much improved, but Reviewer 2 raised some interesting points that likely do not require additional experimentation to address.

We look forward to receiving your revised manuscript.

Kind regards,

Richard A Wilson

Academic Editor

PLOS ONE

Journal Requirements:

Reviewers' comments:

Reviewer's Responses to Questions

**Comments to the Author**

1. If the authors have adequately addressed your comments raised in a previous round of review and you feel that this manuscript is now acceptable for publication, you may indicate that here to bypass the “Comments to the Author” section, enter your conflict of interest statement in the “Confidential to Editor” section, and submit your "Accept" recommendation.

Reviewer #1: All comments have been addressed

Reviewer #2: (No Response)

2. Is the manuscript technically sound, and do the data support the conclusions?

Reviewer #1: Partly

Reviewer #2: Partly

3. Has the statistical analysis been performed appropriately and rigorously? 

Reviewer #1: Yes

Reviewer #2: No

4. Have the authors made all data underlying the findings in their manuscript fully available?

Reviewer #1: Yes

Reviewer #2: Yes

5. Is the manuscript presented in an intelligible fashion and written in standard English?

Reviewer #1: Yes

Reviewer #2: Yes

6. Review Comments to the Author

Reviewer #1: (No Response)

Reviewer #2: The revised version is much-improved. I have however a few concerns, which should be addressed.

1. If feasible a video may complement the Figure 5. I could not follow the concept that mitochondria remain attach to the chloroplast for a minute …... see sentence below from the manuscript.

"The arrow in each image (every 1min:30sec) indicates that the mitochondria localized

FDH1 moves to chloroplasts and attach there approximately for 1 min, and later goes apart from the chloroplast."

2. Figure 6B is an improvement but it lacks statistical analyses.

3. The manuscript does not have any references to any of the publications of 2020 and 2021. There has been a lot known in the last two years. This can easily be fixed. Certain places authors may consider to cite original references rather than reviews, which are old and do not accommodate new work. Instead of using multiple reviews, authors are encouraged to cite the original work so that readers will be able to access those easily.

7. PLOS authors have the option to publish the peer review history of their article (what does this mean?). If published, this will include your full peer review and any attached files.

Reviewer #1: No

Reviewer #2: No

---

## [Author Response · Author response to Decision Letter 1]

17 Feb 2022

Dear Editor, 

Thank you for the valuable comments on our manuscript. We have followed the reviewers’ comments and thoughtfully revised the manuscript. 

Please see below our responses and let us know for any questions. 

Best Regards, 

Seonghee Lee

Journal Requirements:

-> Response: The list of reference was updated and highlighted for the changes. Following the reviewer’s comments, some old references were excluded because there were new publications available. Also, we found that there were too many references cited in the manuscript, and retracted some references that describe duplicated information. 

Reviewers' comments:

Reviewer #1: (No Response)

Reviewer #2: The revised version is much-improved. I have however a few concerns, which should be addressed.

1. If feasible a video may complement the Figure 5. I could not follow the concept that mitochondria remain attach to the chloroplast for a minute …... see sentence below from the manuscript.

"The arrow in each image (every 1min:30sec) indicates that the mitochondria localized FDH1 moves to chloroplasts and attach there approximately for 1 min, and later goes apart from the chloroplast."

- Response: This sentence was changed and also we included a video (supplementary figure) to describe the movement of mitochondria to chloroplast. 

2. Figure 6B is an improvement but it lacks statistical analyses.

Corrected. 

3. The manuscript does not have any references to any of the publications of 2020 and 2021. There has been a lot known in the last two years. This can easily be fixed. Certain places authors may consider to cite original references rather than reviews, which are old and do not accommodate new work. Instead of using multiple reviews, authors are encouraged to cite the original work so that readers will be able to access those easily.

- Response: Reference list was updated. 

---

## [Editor Report · Decision Letter 2]

22 Feb 2022

Functional Role of Formate Dehydrogenase (FDH1) for Host and Nonhost Disease Resistance Against Bacterial Pathogens

PONE-D-20-40925R2

Dear Dr. Lee,

We’re pleased to inform you that your manuscript has been judged scientifically suitable for publication and will be formally accepted for publication once it meets all outstanding technical requirements.

Kind regards,

Richard A Wilson

Academic Editor

PLOS ONE
---

## [Editor Report · Acceptance letter]

20 Apr 2022

PONE-D-20-40925R2 

Functional Role of Formate Dehydrogenase (FDH1) for Host and Nonhost Disease Resistance against Bacterial Pathogens 

Dear Dr. Lee:

I'm pleased to inform you that your manuscript has been deemed suitable for publication in PLOS ONE. Congratulations! Your manuscript is now with our production department. 

Kind regards, 

on behalf of

Dr. Richard A Wilson 

Academic Editor

PLOS ONE